# Evaluation of quality of life and associated factors in patients with intertrochanteric femoral fracture

**Fatemeh Ghasemi**[1], **Seyyed Mokhtar Esmaeilnejad-Ganji**[1], **Alireza Manafi Rasi**[2], **Sina Afzal**[2], **Mojtaba Baroutkoub**[2], **Mehdi Tavassoli**[1] *

1 Department of Orthopedic and Trauma Surgery, Babol University of Medical Sciences, Babol, Iran,
2 Department of Orthopedic and Trauma Surgery, Shahid Beheshti University of Medical Sciences, Tehran, Iran

* Mehdi.tavasolii@gmail.com

## Abstract

### Background

Intertrochanteric fracture is a common injury among the elderly, causing fundamental life-style derangements, influencing the patients' social and psychological functioning. We aimed to study the quality of life (QoL) and its different parameters in patients with this type of injury.

### Materials and methods

In this cross-sectional study, all patients hospitalized with intertrochanteric fracture aged above 50 from 2020 to 2022 at the Shahid Beheshti Hospital in Babol, Iran, were included. Patients were primarily managed surgically and, in exceptional cases non-surgically, were followed up for at least 12 months after receiving treatment. During the follow-up period, patients were dialled and completed a questionnaire to assess the patient's QoL by the 36-item Short Form Health Survey (SF-36). QoL parameters were analyzed based on patients' sex, age, type of treatment, and height of fall causing fracture.

### Result

A total number of 200 patients, including 101 (50.5%) males and 99 (49.5%) females, with a mean age of 74.76±11.36 years (range: 50–99), were included. Regarding the received treatment, 192 (96.0%) patients underwent surgery, and 8 (4.0%) underwent non-surgical treatment. In the study of SF-36 scores, the mean score of male patients was 42.31±14.58, and females scored 37.83±15.35, and the difference was statistically significant (P = 0.04). The mean score of QoL and its subscales among the 50–75 group patients was significantly higher than the 76–99 group (P<0.001). The average score of QoL was considerably higher in patients who had surgery (40.75±14.57) compared to those who had non-surgical treatment (24.30±19.85) (P = 0.01). Patients having a fall from a higher height had higher QoL after treatment.

Committee of Babol University of Medical Sciences (contact via research@mubabol.ac.ir) for researchers who meet the criteria for access to confidential data.

**Funding:** The author(s) received no specific funding for this work.

**Competing interests:** The authors have declared that no competing interests exist.

## Conclusion

This study revealed that patients with an intertrochanteric femoral fracture had poor QoL in all aspects. The overall QoL was significantly higher among male patients, younger patients, those who underwent surgical treatment, and the falls from higher heights. These findings highlight the necessity of long-term follow-up and support in patients with intertrochanteric fractures.

## Introduction

Intertrochanteric fracture is a form of femoral fracture involving both the lesser and greater femoral trochanters and is a common injury among the elderly population [1]. In this kind of the proximal femoral fracture, a fracture line travels obliquely through the trochanters below the femoral neck and does not involve the neck or upper components [2]. Intertrochanteric fractures can develop in the elderly due to minor trauma like falling or in the younger population due to severe trauma like high-energy motor vehicle accident [3]. The incidence rate of intertrochanteric fractures is estimated at 171 per 100,000 population in the United States, bearing nearly 52500 USD economic burden for each patient [4]. Women are more likely to suffer from this form of fracture, which is thought to be due to metabolic and hormonal variations in the bone [5]. Studies from Iran have reported mortality rates of about 30–35% among patients with intertrochanteric fractures who underwent surgical treatment [6, 7].

In the physical examination, the injured limb is generally shortened, the region is inflamed, and movement of the limb is painful [8]. Surgical treatment is the gold standard therapy for intertrochanteric fractures [9]. On the other hand, non-surgical treatment offers a satisfactory improvement rate [8]. Malignancy, failure of the operated device such as protrusion of the device or sinking of the rod in the hip joint, shortening of the limbs, nonunion, and in rare cases, avascular necrosis of the femoral head could be associated with intertrochanteric fractures [10, 11]. Intertrochanteric fractures cause a lot of morbidity and mortality, so knowing about post-operative complications can help patients have a better quality of life (QoL) [7].

Today, there is a significant focus on the QoL of patients with trauma and fractures since the evidence shows that the QoL following trauma is much lower than before the event [11]. Thus, patients should be followed up after surgical treatment for intertrochanteric fracture since the altered QoL remains low [6, 12]. Patients' poor QoL after surgery may be due to the lack of follow-up after discharge, extensive medical expenditures, inadequate rehabilitation, or a lack of social, physical, and emotional support [13–15]. Therefore, in this study, we aimed to assess the QoL in patients undergoing surgery for an intertrochanteric fracture to help clinicians improve the care provided for such patients with a final goal of enhancing surgical and patient-reported outcomes.

## Material and methods

### Study design and population

In this cross-sectional study, all patients hospitalized with intertrochanteric fracture from 07/04/2020 to 07/04/2022 at the Shahid Beheshti Hospital in Babol, Iran, were included. Inclusion criteria were having an intertrochanteric fracture and being above 50 years of age at the time of hospitalization or surgery. Exclusion criteria from the study were the duration between fracture and admission to the hospital being more than three weeks as a longer time from injury to

surgery increases the probability of complications and less fortune post-operative outcomes [16], established dementia, if the cause of fracture was a car accident or fall from a height of more than two meters as such injuries are different from the usual intertrochanteric fractures which are assumed to be a senile fracture in the geriatric population [16], and patients with a severe underlying condition.

## Patient management

The patient's unsatisfactory general state, the inability to perform anesthesia, the delay in treatment (for more than three weeks from the fracture), and the patient's preference to not go under surgery were all reasons for non-surgical treatment. Otherwise, the patients were treated surgically. Regarding the patients who underwent the surgical management, on average, 3–5 days after surgery, patients were discharged with physiotherapy training and appropriate recommendations. In contrast, non-surgical patients were discharged after the necessary training, cane preparation, or stretching to continue treatment at home. During hospitalization, these therapies lasted an average of at least three days. Patients were contacted at least 12 months after being admitted to the hospital or having surgery.

## Data collection and follow-up

During the follow-up period, patients were dialled and completed a questionnaire to assess the patients' QoL by the 36-item Short Form Health Survey (SF-36). This questionnaire contains 36 questions and eight subscales, each with two to ten items. The eight subscales of this questionnaire are physical function (PF), role disorder due to physical health (RP), role disorder due to emotional health (RE), energy/fatigue (EF), emotional well-being (EW), social function (SF), Pain (P) and general health (GH) [17]. In this questionnaire, a lower score indicates a lower quality of life and vice versa. Also, the SF-36 tool has been previously translated and validated for Iranian patients, and we used that version for this study [18]. QoL parameters were analyzed based on patients' sex (male/female), age group (50–75 and 76–99 years old), type of treatment (surgical/non-surgical), and height of fall causing fracture (level height/1 meter height/2 meters height).

## Statistical analysis

Categorical variables were summarized in frequency and percentage, and the quantitative variables were described by mean and standard deviation or median and interquartile range based on normality. To evaluate the statistical associations between variables, parametric (T-test) and non-parametric (Mann-Whitney and Kruskal-Wallis) tests were used based on normality assumptions of variables. SPSS version 25 was used for all data analysis. Two-sided P-values less than 0.05 were investigated and set as the level of statistical significance.

## Ethical considerations

This study was conducted per the statement of ethical principles of the Declaration of Helsinki. 'The study protocol was reviewed and approved by the research ethics committee of the School of Medicine at Babol University of Medical Sciences, Babol, Iran (code: IR.MUBABOL. REC.1399.007). All patients provided written informed consent to participate in the study and for the publication of its results. During data collection and analysis, appropriate labelling of data was used to anonymize the data and save patients' privacy.

**Table 1. Comparison of quality of life subscales in male and female patients with intertrochanteric fracture.**

| Quality of Life | Male (mean±SD) | Female (mean±SD) | P-value |
|---|---|---|---|
| Physical function | 22.52±13.74 | 19.39±10.76 | 0.22 |
| Role disorder due to physical health | 26.73±27.44 | 22.72±25.02 | 0.33 |
| Role disorder due to emotional health | 56.43±39.08 | 45.45±41.36 | 0.06 |
| Energy / Fatigue | 48.86±10.31 | 44.94±12.64 | 0.09 |
| Emotional well-being | 55.56±10.44 | 52.24±13.10 | 0.06 |
| Social function | 35.27±19.95 | 35.95±21.68 | 0.38 |
| Pain | 58.88±23.48 | 54.39±24.39 | 0.22 |
| General health | 34.20±14.96 | 30.55±15.46 | 0.07 |
| Physical health | 35.58±16.16 | 31.76±15.34 | 0.11 |
| Mental health | 49.03±15.74 | 43.90±17.74 | 0.06 |
| Total | 42.31±14.58 | 37.83±15.35 | 0.04 |

## Results

A total number of 200 patients, including 101 (50.5%) males and 99 (49.5%) females, were included in this study. The mean age of patients was 74.76 ± 11.36 years (range: 50–99). Ninety-four (47.0%) patients were in the 50–75 age range, and 106 (53.0%) were in the age group of 76–99 years age range. Inspecting the height leading to fall and intertrochanteric fracture, 157 (78.5%) patients fell from the same height, 35 (17.5%) fell from a height of about two meters, and 8 (4.0%) fell from a height of one meter. Regarding the received treatment, 192 (96.0%) patients underwent surgery, and 8 (4.0%) patients underwent non-surgical treatment for various reasons.

In the study of SF-36 scores, the mean score of male patients was 42.31±14.58, and females scored 37.83±15.35, and the difference was statistically significant (P = 0.04). However, in QoL subscales, no significant differences were observed between male and female patients with intertrochanteric fractures (Table 1).

In the study of subscales of QoL of patients in the age groups of 50–75 and 76–99, it was found that the mean score of QoL and its subscales among the patients of the 50–75 group was significantly higher than the 76–99 group, as the mean total score of the first group was 46.70 ±11.79 compared to the second group which was 34.23±15.33 (P<0.001) (Table 2).

The average score of QoL was considerably higher in patients who had surgery (40.75 ±14.57) compared to those who had non-surgical treatment (24.30±19.85), reflecting a

**Table 2. Comparison of quality of life subscales in patients with intertrochanteric fracture by age group.**

| Quality of Life | 50–75 age group (mean±SD) | 76–99 age group (mean±SD) | P-value |
|---|---|---|---|
| Physical function | 26.70±9.26 | 15.86±12.70 | <0.001 |
| Role disorder due to physical health | 34.84±25.45 | 15.80±23.73 | <0.001 |
| Role disorder due to emotional health | 58.86±36.06 | 44.02±43.04 | 0.005 |
| Energy / Fatigue | 51.01±10.06 | 43.30±11.82 | <0.001 |
| Emotional well-being | 57.95±9.84 | 50.33±12.48 | <0.001 |
| Social function | 40.15±19.56 | 28.77±20.49 | <0.001 |
| Pain | 65.29±23.71 | 49.00±21.59 | <0.001 |
| General health | 38.82±12.19 | 26.69±15.53 | <0.001 |
| Physical health | 41.41±13.18 | 25.58±14.86 | <0.001 |
| Mental health | 51.99±13.36 | 41.61±18.25 | <0.001 |
| Total | 46.70±11.79 | 34.23±15.33 | <0.001 |

**Table 3. Comparison of quality of life subscales in patients with intertrochanteric fracture based on type of treatment.**

| Quality of Life | Surgical Treatment (mean ±SD) | Non-surgical Treatment (mean ±SD) | P-value |
|---|---|---|---|
| Physical function | 21.14±11.37 | 16.87±29.02 | 0.03 |
| Role disorder due to physical health | 25.52±26.33 | 6.25±17.67 | 0.03 |
| Role disorder due to emotional health | 52.43±40.28 | 16.66±30.86 | 0.01 |
| Energy / Fatigue | 47.57±10.98 | 31.25±16.85 | 0.004 |
| Emotional well-being | 54.54±11.19 | 39.00±18.85 | 0.01 |
| Social function | 34.57±20.51 | 23.43±26.25 | 0.11 |
| Pain | 57.42±23.87 | 38.43±20.17 | 0.02 |
| General health | 32.81±14.99 | 22.50±19.82 | 0.11 |
| Physical health | 34.22±15.49 | 21.01±19.82 | 0.01 |
| Mental health | 47.28±16.33 | 27.58±20.74 | 0.01 |
| Total | 40.75±14.57 | 24.30±19.85 | 0.01 |

statistically significant difference (P = 0.01), with similar significant patterns in subscales except for the two subscales of social function and general (Table 3).

Categorized based on the height of the fall causing intertrochanteric, the total QoL score and subscales varied in three categories of height among the included sample of patients in this study (Table 4).

## Discussion

This study investigated the QoL and its different parameters among patients with intertrochanteric fractures. The main findings derived from the recruited sample of patients in this study were significantly higher QoL among male patients with this kind of fracture, younger patients, those who underwent surgical treatment, and falls from higher heights. Variations in subscales of QoL were also considerable in different stratifications of patients investigated in this study reflecting varying aspects of QoL affected by such injuries.

The most notable finding of this study was that intertrochanteric femoral fractures cause derangements in patient QoL, even when there were no complications and in cases with full

**Table 4. A comparison of quality of life subscales in patients with an intertrochanteric fracture based on the height of the fall that caused the fracture.**

| Quality of Life | Level height (mean±SD) | Height 1 meter (mean±SD) | Height 2 meters (mean±SD) | P-value |
|---|---|---|---|---|
| Physical function | 19.39±12.06[a*] | 26.00±11.74[b] | 30.00±14.14[bc] | 0.001 |
| Role disorder due to physical health | 21.49±25.39[a] | 28.57±27.34[b] | 38.12±20.86[ba] | 0.003 |
| Role disorder due to emotional health | 49.46±41.27 | 56.19±35.94 | 58.33±46.29 | 0.58 |
| Energy / Fatigue | 46.21±11.34[a] | 50.85±11.57[b] | 51.37±12.37[ab] | 0.006 |
| Emotional well-being | 52.71±12.10[a] | 54.31±8.70[b] | 59.00±15.11[ab] | 0.005 |
| Social function | 33.51±20.94 | 30.14±20.89 | 32.81±18.82 | 0.55 |
| Pain | 54.69±24.18 | 64.00±22.53 | 63.12±21.07 | 0.09 |
| General health | 31.56±15.00 | 31.42±16.02 | 36.25±16.63 | 0.21 |
| Physical health | 31.78±15.62[a] | 38.25±14.74[b] | 41.12±15.58[ab] | 0.004 |
| Mental health | 45.47±17.11 | 51.12±15.31 | 46.13±18.44 | 0.27 |
| Total | 38.63±15.00[a] | 46.18±13.80[b] | 48.12±15.43[ab] | 0.03 |

*Different letters in each row indicate a significant difference between the two groups at the level of α = 0.05 using Tukey test.

healing achieving. The QoL of trauma patients and those with fractures and bone and soft tissue injuries was not well studied in Iran, and most of the available studies are from Western countries [19]. However, regarding intertrochanteric fractures, some studies from Iran assessed the outcomes of treatment of these injuries and the QoL of patients having the injury [6, 7, 15]. In a study in Iran on 385 patients aged≥60 years old who had intertrochanteric fractures and were treated surgically by open reduction and internal fixation by dynamic hip screw, the mortality rate after surgery was 36.9% with higher figures among males (54.9%) compared to females (41.9%), and only 33.5% of the patients had a good score of QoL based on the Modified Harris Hip Score, concluding that QoL even postoperatively remains low in this patients and needs appropriate follow up to improve the treatment outcomes and patient-reported QoL [7]. In another study on 110 patients with intertrochanteric fractures from Iran who were treated by dynamic hip screw and the QoL was assessed by the SF-36, similar to our study, reported no association between the overall score of QoL and its physical and mental subscales with patients' sex; however, mortality after surgery and QoL was negatively associated with patients' age, which their findings were comparable to our results [6]. One study from Iran on 81 patients with intertrochanteric fracture who were surgically treated and the QoL was assessed by the SF-36 tool reported the mean QoL score of 48.5±17.7 among enrolled patients, and further adjusted analysis found statistically significant associations between patient age, economic status, the fracture to surgery interval, and the QoL (P<0.001), concluding the overall QoL of these patients being moderate to low and highlighted the importance of appropriate follow-up and psychological support, especially in older individuals with intertrochanteric fractures [15].

Studies from other centers worldwide also have investigated the QoL parameters in patients with intertrochanteric fractures [14, 20–22]. In a study of 86 patients aged>60 years old with isolated and unstable intertrochanteric fractures and were in two groups of intramedullary nailing and antirotator proximal femoral nail implant (42 cases) and cementless bipolar hemiarthroplasty (44 cases) and were followed for 24 months and various tools assessed QoL, the results were indicative of better social functioning and mobility scores in the internal fixation group with significantly better improvement rates in this group over time (P<0.01) [14]. A long-term cohort study of 200 cases with intertrochanteric fracture and the healthy age- and sex-matched controls found that the QoL of patients with this injury had significantly lower QoL, especially regarding activities in daily living functions compared to the healthy controls [20]. One investigation on 117 patients with unstable intertrochanteric fractures treated in two groups of the proximal femoral nail (66 cases), hemiarthroplasty (42 cases), or total hip arthroplasty (9 cases), found the proximal femoral nailing the superior method in terms of patient mobilization and QoL parameters [21].

In an interventional trial on 482 cases of senile osteoporotic intertrochanteric fracture of femur who were treated by proximal femoral intramedullary fixation and then received zoledronic acid or calcium carbonate/Vitamin D3, the results showed that prescription of zoledronic acid postoperatively improves both bone metabolism and QoL of patients making this agent a viable option to improve the QoL after surgery and facilitate bone healing [23]. One retrospective cohort study on elderly patients with intertrochanteric fractures in terms of QoL showed that age is an independent factor in achieving better outcomes that should be considered in patient management [22], which was similar to our findings.

Other comparative research on QoL on different kinds of injuries also exists. In a study by Hagino et al., it was discovered that patients with pelvic fractures, particularly intertrochanteric fractures, have a lower QoL than those with other types of fractures [24]. The QoL in patients decreased slightly in all eight subscales of the SF-36 tool in a study by Fanian et al., with the most significant decrease in the areas of limited activity due to physical injury and

physical function, which is consistent with the findings of the current study [12]. Because this sort of fracture has a detrimental impact on patients' health and QoL in general, fractures in elderly patients have an even more significant impact [4]. The amplification of all of the situations mentioned above would be beyond comprehension due to the absence of good physical fitness and minerals in the body of the elderly population [20].

In this study, female patients had a lower QoL than male patients. Given males' better physical status of bones, the issues created by many pregnancies and births in women, and their direct influence on physical ability, females' lower QoL in this study could be justified. Also, patients of younger ages had a greater general QoL and its subscales in this study. Factors such as chronic illness, bone resorption, inability to do personal duties, and diminished control over the environment appear to impair and reduce the QoL in older individuals [7].

Another finding of the current study was the higher QoL among patients with falls from higher heights compared to those from a level height or a lower height. This finding was interesting, and we believe it is mainly due to the nature of intertrochanteric fractures happening at different heights as it is somehow related to the age and pre-trauma and fracture status of the patient, as patients having fracture from higher heights compared to those having this injury at the level height usually are at a younger age and can climb for a higher height for their activities. Therefore, it is probable that such patients with better activity and health status before fracture report a higher QoL after surgery and treatment of intertrochanteric fracture. Further research on the associated factors to QoL after intertrochanteric fractures is needed to reveal the exact reasons behind such differences in findings among affected patients.

This study had some limitations. The limited sample size and follow-up period were among the top limitations. The lack of records on complication rates and outcomes after surgery was another limitation. Also, the QoL was assessed only by one tool that could be expanded in future studies to achieve the best estimations on the affected individuals by intertrochanteric fractures. Another limitation of this study was the lack of sample size and quality sufficiency to conduct further complex statistical analysis, including the development of predictive models, which need further expansion of data collection and quality improvement in future to enable investigators to conduct such analyses.

In contrast, this study had some strengths, too. Including various factors in study analysis was the main strength of this study to evaluate the QoL by several stratifications. Including two groups of patients with and without surgical treatment and comparing QoL was the other study strength. This study's findings could be used to improve the QoL of elderly population affected by this injury and similar research in the field to benefit the patients.

## Conclusion

The findings of this study revealed that patients with an intertrochanteric femoral fracture had poor QoL in all aspects. The overall QoL was significantly higher among male patients with this kind of fracture, younger patients, those who underwent surgical treatment, and those who fell from higher heights. These findings highlight the necessity of long-term follow-up in patients with intertrochanteric fractures besides the medical and psychosocial support, to improve the treatment outcomes and QoL.

## Supporting information

**S1 File.**
(PDF)

## Acknowledgments

The authors would like to thank all who contributed to this study.

## Author Contributions

**Conceptualization:** Seyyed Mokhtar Esmaeilnejad-Ganji, Mehdi Tavassoli.

**Data curation:** Fatemeh Ghasemi.

**Methodology:** Alireza Manafi Rasi.

**Writing – original draft:** Mojtaba Baroutkoub.

**Writing – review & editing:** Sina Afzal.

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
