## [Decision Letter · Decision Letter 0]

20 Sep 2023

PONE-D-23-23806Evaluation of quality of life and associated factors in patients with intertrochanteric femoral fracturePLOS ONE

Dear Dr. Tavasoli,

Thank you for submitting your manuscript to PLOS ONE. After careful consideration, we feel that it has merit but does not fully meet PLOS ONE’s publication criteria as it currently stands. Therefore, we invite you to submit a revised version of the manuscript that addresses the points raised during the review process.

We look forward to receiving your revised manuscript.

Kind regards,

Satabdi Mitra, M.D(Community Medicine )

Academic Editor

PLOS ONE

Additional Editor Comments:

We are thankful for submission of the valuable work. Hereby we are sending you the suggestions from the reviewers for your further consideration.

Reviewers' comments:

Reviewer's Responses to Questions

**Comments to the Author**

1. Is the manuscript technically sound, and do the data support the conclusions?

Reviewer #1: Yes

Reviewer #2: Yes

2. Has the statistical analysis been performed appropriately and rigorously? 

Reviewer #1: No

Reviewer #2: Yes

3. Have the authors made all data underlying the findings in their manuscript fully available?

Reviewer #1: Yes

Reviewer #2: Yes

4. Is the manuscript presented in an intelligible fashion and written in standard English?

Reviewer #1: Yes

Reviewer #2: Yes

5. Review Comments to the Author

Reviewer #1: Comment 1:

Page:2 Line: 38: “Cohort study” This is a cross sectional study and cannot be called a cohort study as there are no groups (exposed and non-exposed) which have been studied before the onset of exposure (Inter trochanteric Fracture). Hence, authors are requested to remove the word “Cohort”.

Comment 2:

Page:4 Line:94: Same comment as above. Kindly remove the word “Cohort”.

Comment 3:

Page:4 Line: 98-99: Authors can add details or references for the basis of selection of exclusion criteria such as “admission to the hospital being more than three weeks,”, “if the cause of fracture was car accident or fall from a height of more than two meters,”.

Comment 4:

Overall, the authors have worked well on the manuscript. However, considering the designations of the authors, they can go for higher levels of statistical analysis such as deriving of predictive models using regression analysis to assess if the factors such as age group, height of fall, gender, type of treatment opted by the patients etc. contribute significantly to the total SF 36 score or not.

Reviewer #2: The authors can use any software like grammarly for improving the clarity of statements and tenses.

The discussion part needs to discuss on the higher qol among patients with higher height fractures

The tool can be included in the Annexure

6. PLOS authors have the option to publish the peer review history of their article (what does this mean?). If published, this will include your full peer review and any attached files.

Reviewer #1: **Yes: **Dr K Md Shoyaib

Reviewer #2: No

---

## [Author Response · Author response to Decision Letter 0]

12 Oct 2023

Dear Reviewers,

We would like to thank you for your kind consideration and efforts. We tried to revise the manuscript based on the received comments and we believe this could majorly enhance the prepared draft. We hope the amendments and responses to the raised queries could meet the expectations and resolve the shortcomings of this manuscript. Below, please find the point-by-point response to each provided comment.

Reviewer #1: 

Comment 1:

Page:2 Line: 38: “Cohort study” This is a cross sectional study and cannot be called a cohort study as there are no groups (exposed and non-exposed) which have been studied before the onset of exposure (Inter trochanteric Fracture). Hence, authors are requested to remove the word “Cohort”.

Response: We sincerely thank the reviewer for this wise comment. We agree with the reviewer on this comment regarding the study design.¬ We revised the study design to cross-sectional as recommended through the abstract and main text (lines 28 and 89).

Comment 2:

Page:4 Line:94: Same comment as above. Kindly remove the word “Cohort”.

Response: We revised the methods section accordingly (line 89).

Comment 3:

Page:4 Line: 98-99: Authors can add details or references for the basis of selection of exclusion criteria such as “admission to the hospital being more than three weeks,”, “if the cause of fracture was car accident or fall from a height of more than two meters,”.

Response: We thank the reviewer for highlighting this point in the methods of our study. Selection criteria to exclude such patients were based on the well-known reference of fractures in orthopedics surgery, the Rockwood and Green's Fractures in Adults which is the reference for the management of fractures in adults in our center. Based on this textbook, high-energy traumas like car accident or fall from a height of more than two meters that could cause intertrochanteric fractures are different in nature from the senile fracture of the intertrochanteric area that we intended to assess in this study. Also, regarding the other mentioned exclusion criteria, since the most important predictor of the post-operative outcomes in intertrochanteric fractures is the time period between injury and surgery, again based on our reference textbook, patients with such time more than three weeks are at higher risk of complications and less fortunate surgical outcomes. Therefore, this was the other exclusion criteria in this study. For the reference of the readership of this paper, we added some explanation and the citation to the textbook in the related part of the methods section which could be found in lines 92-98.

Comment 4:

Overall, the authors have worked well on the manuscript. However, considering the designations of the authors, they can go for higher levels of statistical analysis such as deriving of predictive models using regression analysis to assess if the factors such as age group, height of fall, gender, type of treatment opted by the patients etc. contribute significantly to the total SF 36 score or not.

Response: With many thanks for this comment, although there is a potential to conduct further complex analysis on the collected data to develop predictive models in this study, we believe the limited sample size and quality of data puts a major challenge against development of such models in this study. So, we preferred to stick to the descriptive and primary analytic data presentation in this submission. We tried to cover this issue as a limitation at the end of the manuscript. Please find the added limitation in lines 263-266. We truly appreciate the kind consideration of the reviewer on this notion.

Reviewer #2: 

The authors can use any software like grammarly for improving the clarity of statements and tenses.

Response: We genuinely thank the reviewer for the comment on the quality of language and grammar in this submission. We used the help of a professional medical text editor in this revision to resolve the grammar errors through the text. 

The discussion part needs to discuss on the higher qol among patients with higher height fractures

Response: Thanks for this bright comment. This point is a very interesting issue raised by the reviewer and we believe there is a possible reason behind this finding as patients experiencing intertrochanteric fractures from higher heights usually have a better pre-injury health status, so they have a higher quality post-injury period and experience better rehabilitation. We tried to expand this point in-depth in a paragraph in the discussion section to provide a justification for the finding and benefit the readership of this paper. The added parts could be found in lines 249-258.

The tool can be included in the Annexure

Response: With many thanks for this beneficial comment, we added the SF-36 tool as a supplementary to this manuscript in this revision, retrieved from the source we adopted (https://www.rand.org/health-care/surveys_tools/mos/36-item-short-form/survey-instrument.html).

Editor comments on manuscript:

Response: With many thanks for the several comments left on the file of submission, we addressed them all to improve the quality of manuscript. Besides, a comprehensive language and grammar edit of all parts of the manuscript was done by a professional medical text editor in this revision to improve the readability of the submission. The edited parts of the manuscript could be found in lines 130-137, 163, 180, 186, 189, 191, 249-258.

---

## [Editor Report · Decision Letter 1]

17 Oct 2023

Evaluation of quality of life and associated factors in patients with intertrochanteric femoral fracture

PONE-D-23-23806R1

Dear Dr. Tavasoli,

We’re pleased to inform you that your manuscript has been judged scientifically suitable for publication and will be formally accepted for publication once it meets all outstanding technical requirements.

Kind regards,

Satabdi Mitra, M.D(Community Medicine )

Academic Editor

PLOS ONE
---

## [Editor Report · Acceptance letter]

13 Nov 2023

PONE-D-23-23806R1 

Evaluation of quality of life and associated factors in patients with intertrochanteric femoral fracture 

Dear Dr. Tavasoli:

I'm pleased to inform you that your manuscript has been deemed suitable for publication in PLOS ONE. Congratulations! Your manuscript is now with our production department. 

Kind regards, 

on behalf of

Dr Satabdi Mitra 

Academic Editor

PLOS ONE